# Crystal structure and chemical inhibition of essential schistosome host-interactive virulence factor carbonic anhydrase SmCA

Akram A. Da'dara[1], Andrea Angeli[2], Marta Ferraroni[3], Claudiu T. Supuran[2] & Patrick J. Skelly [1]

The intravascular parasitic worm *Schistosoma mansoni* is a causative agent of schistosomiasis, a disease of great global public health significance. Here we identify an α-carbonic anhydrase (SmCA) that is expressed at the schistosome surface as determined by activity assays and immunofluorescence/immunogold localization. Suppressing SmCA expression by RNAi significantly impairs the ability of larval parasites to infect mice, validating SmCA as a rational drug target. Purified, recombinant SmCA possesses extremely rapid $CO_2$ hydration kinetics ($k_{cat}$: $1.2 \times 10^6$ s$^{-1}$; $k_{cat}/K_m$: $1.3 \times 10^8$ M$^{-1}$s$^{-1}$). The enzyme's crystal structure was determined at 1.75 Å resolution and a collection of sulfonamides and anions were tested for their ability to impede rSmCA action. Several compounds (phenylarsonic acid, phenylbaronic acid, sulfamide) exhibited favorable $K_i$s for SmCA versus two human isoforms. Such selective rSmCA inhibitors could form the basis of urgently needed new drugs that block essential schistosome metabolism, blunt parasite virulence and debilitate these important global pathogens.

[1] Department of Infectious Disease and Global Health, Cummings School of Veterinary Medicine, Tufts University, North Grafton, MA, USA. [2] Sezione di Scienze Farmaceutiche e Nutraceutiche, Dipartimento Neurofarba, Università degli Studi di Firenze, Via U. Schiff 6, 50019 Sesto Fiorentino, Florence, Italy. [3] Dipartimento di Chimica " Ugo Schiff", Università di Firenze, Via della Lastruccia 3, 50019 Sesto Fiorentino, Florence, Italy. Correspondence and requests for materials should be addressed to P.J.S. (email: Patrick.Skelly@Tufts.edu)

Schistosomiasis is a disease of enormous public health significance. It is caused by intravascular parasitic worms commonly called blood flukes and ranks among the most important infectious diseases worldwide[1]. Over 200 million people are currently infected with these parasites[2]. Conservatively, mortality is put at >280,000 deaths per year, with tens of millions having chronic morbidity[3,4]. Extensive pathological changes associated with schistosome infection can affect multiple organ systems, including the liver, spleen, urogenital, and gastro-intestinal tracts, especially in individuals with longstanding or heavy infections[5]. The condition can be particularly devastating for growing children, resulting in stunted physical and cognitive development[4]. In pregnant women, infection can result in poor fetal outcomes[3]. Most human infections are caused by *Schistosoma mansoni*, *S. haematobium* or *S. japonicum*. Over 800 million people live at risk of infection by these parasites[4].

Currently, there is no vaccine available to prevent schistosome infection. Control is essentially limited to treatment with the drug praziquantel (PZQ). While safe and effective, this drug does not kill juvenile parasites[6–8] In addition, it has been in wide use for over 30 years and a series of recent laboratory studies and clinical trials have raised concern about the development of tolerance and/or resistance to it[9–12]. The World Health Organization, recognizing that continued reliance on a single curative drug is risky, has—along with others in the scientific community—called for the urgent development of new interventions for the prevention and cure of schistosomiasis, one aim of this work[13–15].

Adult *Schistosoma mansoni* worms, living largely as male-female pairs, are mostly found within the mesenteric venous plexus of their mammalian hosts where they can survive for many years. The intravascular worms are covered by a structurally unique double-lipid bilayer whose protein composition has been investigated using proteomics[16,17]. Proteins identified include nutrient transporters, receptors, and enzymes and several of unknown function[18–22]. In these proteomic studies, a putative carbonic anhydrase (CA) was identified;[18,19] this enzyme is the focus of this work and is shown here to be essential for the worms to establish robust infection in experimental animals.

Carbonic anhydrases (EC 4.2.1.1) are ubiquitous zinc metalloenzymes, present in all kingdoms of life, and encoded by at least seven distinct, evolutionarily unrelated gene families (designated α, β, γ, δ, ζ, η, and θ[23,24]). The α-CAs predominate among animals and are the only CA gene family found in vertebrates. In humans, a total of 15 CA isoforms have been identified and these differ in their cellular localization, concentration, and catalytic efficiency. Most are cytosolic (e.g., hCA I and hCA II); others are membrane-bound (e.g., hCA IV)[24,25]. Carbonic anhydrases are best known for their ability to rapidly catalyze the reversible hydration of carbon dioxide to form bicarbonate and protons according to the reaction: $CO_2 + H_2O \leftrightarrow H_2CO_3^- + H^+$. Thus, the enzymes play crucial roles in regulating pH and in the transport of $CO_2$/bicarbonate between metabolizing tissues, which impact ammonia transport, bone resorption, gastric acidity, muscle contraction, gluconeogenesis, renal acidification, and brain development[25,26]. Carbonic anhydrases have additionally been shown to catalyze a variety of other reactions e.g., some display esterase activity, whose physiological significance is unclear[25,27,28].

The putative CA identified by proteomics in the *S. mansoni* tegument (skin)—here designated SmCA—was previously shown to be available for surface biotinylation in living adult worms, highlighting its exposed nature at the parasite surface[18]. In addition, SmCA was released from live worms following their treatment with phosphatidylinositol phospholipase C (PIPLC), suggesting that SmCA is linked to the external parasite surface membrane via a glycosylphosphatidylinositol (GPI) anchor[19].

The presence of a protein of this nature had been earlier implied by the detection of non-specific esterase activity in the schistosome tegument using histochemistry[29] and, as mentioned, CAs can display such activity.

We show here that SmCA is active at the host-parasite interface of the intravascular life stages. Suppressing SmCA gene expression using RNAi impairs the ability of larval schistosomes to establish infection in vivo, revealing this molecule to be important for parasite virulence. This result strongly suggests that chemical inhibition of the enzyme by drug treatment will, by mimicking the RNAi effect, debilitate the worms and curtail the infection. Carbonic anhydrase enzymes are druggable targets[30,31] and many members of this protein family from vertebrates to bacteria (and including other parasitic worms) are known drug targets[32–35].

Here, as a first step towards developing SmCA as a therapeutic target for schistosomiasis, we generate SmCA in recombinant form, determine its crystal structure and identify compounds that preferentially block the activity of the parasite enzyme compared to two human CA isoforms. Such compounds represent novel leads towards the development of new, clinically useful, anti-schistosome therapeutics.

## Results

**SmCA cloning and sequence analysis**. To identify the entire SmCA cDNA, EST sequence information derived from proteomic analysis of the tegumental membranes[18,19] was first used to examine the *S. mansoni* genome database (version 3, http://www.sanger.ac.uk/Projects/S_mansoni/). In this manner, the putative SmCA gene was identified and, as outlined in Methods, genomic sequence information was used to design predicted 5′ and 3′oligonucleotides that were then used to generate the complete SmCA cDNA by PCR. The cDNA potentially encodes the 323-amino acid SmCA protein (GenBank accession no. MK611932) with a predicted molecular mass of 36,831 Da and a predicted pI of 6.06. Supplementary fig. 1A shows an alignment of SmCA with other members of this protein family generated using ClustalW. The schistosome protein falls within the α-CA family and shows high sequence identity (80%) with its homolog from *S. haematobium* (ShCA), 62% identity with its homolog from *S. japonicum* (SjCA), and lesser identity (34%) with human CA isoform IV (HsCA-IV). Relationships between selected CAs from a variety of animals are depicted in a phylogenetic tree generated by neighbor joining with Accelrys Gene software (supplementary fig. 1B). SmCA is predicted to have an amino terminal, 20 amino acid signal sequence ($^1$MTYQWLIGIQISLLFVNCIC$^{20}$) as assessed at www.cbs.dtu.dk/services/SignalP-3.0/. Conserved zinc-binding histidine residues are present ($H^{117}$, $H^{119}$, $H^{142}$). The positions of these histidines follow the pattern for α-CAs, which have a catalytic zinc coordinated by three such residues (at positions: x, x + 2, and x + 25, where x = 117 in SmCA). Four other highly conserved, critical active site residues ($H^{88}$, $Q^{115}$, $E^{129}$, and $T^{231}$) are evident. The arginine residue at position 295 ($R^{295}$) is considered to be the best potential GPI-modification site. Several potential N- glycosylation sites, but no O-glycosylation sites, are predicted in SmCA (at http://www.cbs.dtu.dk/services/NetNGlyc/).

The complete SmCA cDNA was used to interrogate both the *S. haematobium* and *S. japonicum* genome databases (schistoDB.net) in order to annotate existing partial tegumental CA sequences from these species and it is these updated sequences that are presented in supplementary fig. 1A. In addition, using the available *S. mansoni* genome sequence at GeneDB.org, the SmCA gene was identified as being located on chromosome 6 (at position 6,661,661–6,670,674); the gene has 5 exons and extends over ~10 kb (supplementary fig. 1C).

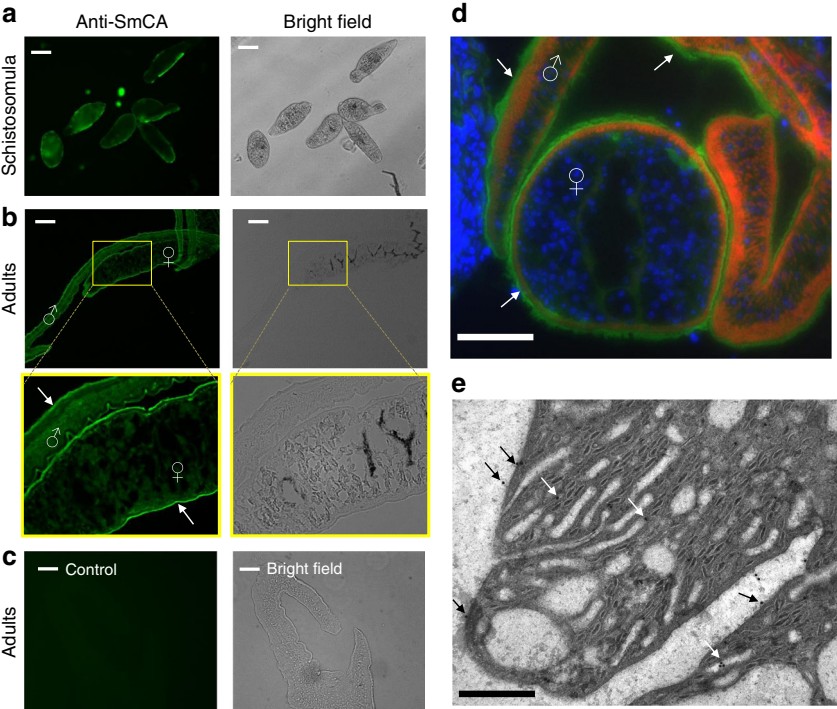

**Fig. 1** Immunolocalization of SmCA in schistosomula (**a**) and in adult males and females (**b**, **d**). No staining is seen in controls (treated with secondary antibody alone, **c**). Bright field images are depicted in **a**–**c** (as indicated) as well as images of parasites incubated with anti-SmCA antibodies (**a**, **b**) or control (**c**), secondary antibody alone. **a** shows intact, fixed schistosomula. **b**–**d** show sections of adult worms. Regions bounded by yellow boxes in **b** are enlarged to highlight the strong staining seen at the periphery of both male and female worms (arrows). **d** is an adult pair in situ in the vasculature of an infected mouse stained with anti-SmCA antibodies (green), DAPI (blue, staining nuclei), and phalloidin (red, staining actin in muscle). Strong staining of SmCA in the tegument (green) of both male and female is indicated by arrows. **e** Electron micrographs of the adult tegument showing immunogold labeling of SmCA. Black arrows indicate localization of gold particles in the tegument at the host-parasite interface and white arrows indicate gold particles in tegumental pits (which connect to the exterior). Scale bar in **a** = 50 μm; in **b** and **c** = 250 μm, in **d** = 200 μm and in **e** = 0.5 μm

**SmCA localization in schistosomula and adult worms**. Figure 1a shows a bright field image of a collection of 3-day cultured schistosomula alongside an image depicting this group staining strongly with anti-SmCA antibody (with especially bright staining at the periphery of these juvenile worms). Figure 1b shows bright field images of adult male and female worms in cross section alongside images showing strong anti-SmCA staining in the worms, particularly in the periphery. Yellow boxes bound regions of the adult worm sections that are shown in higher magnification just beneath—highlighting again the strong tegumental staining evident in both sexes (arrows). Figure 1c shows the absence of background staining in a control worm section when tested with secondary antibody alone. Figure 1d illustrates a cross section of an adult worm couple in the vasculature, with SmCA staining in green (FITC-labeled), nuclei staining in blue (DAPI-labeled) and actin staining in red (phalloidin-labeled). Bright green SmCA staining exterior to the muscle tissue (stained red) reveals the predominant tegumental localization of this protein in both male and female worms (arrows). Blue staining to the left in this image represents nuclei of the surrounding murine tissue. Localization of SmCA by immunogold electron microscopy (Fig. 1e) confirms that the protein is distributed on the host-interactive tegumental membranes. Black arrows in Fig. 1e point to some of the tegumental immunogold particles at the host-parasite interface; white arrows point to staining in tegumental pits (which also connect to the surface).

To assess whether SmCA was functional on the external surface of the parasites, groups of live parasites were incubated in assay buffer and enzyme activity was monitored. Figure 2a shows that groups of living schistosomula are capable of cleaving the CA substrate para-nitrophenyl acetate (p-NPA) as seen by the rise in $OD_{405}$ with time. As expected, increasing the number of schistosomula from 250 to 500 correspondingly increases the activity detected. As seen in Fig. 2b, individual, live, adult male and female worms in assay buffer likewise exhibit SmCA activity, with the larger males displaying greater activity compared to their female counterparts.

**Suppression of SmCA gene expression**. The SmCA gene is expressed in all schistosome life stages examined (egg, cercaria, schistosomulum, adult female, and adult male) with the highest relative expression level seen in the adult males (supplementary fig. 2). Gene expression was suppressed in adult male worms in vitro by introducing a target-specific siRNA using electroporation. Figure 3a shows the specific and robust suppression of SmCA gene expression (by ~80%) as determined by RT-qPCR and measured 2 days after siRNA treatment. Western blot analysis demonstrates that this treatment also results in substantial suppression of SmCA protein production compared to controls, measured 7 days post siRNA treatment (Fig. 3b). Notably lower levels of SmCA protein are detected in extracts from SmCA siRNA-treated parasites (left lane, Fig. 3b, arrow) versus controls (middle and right lanes, Fig. 3b). The lower panel in Fig. 3b shows the same blot re-probed with an irrelevant antibody to illustrate that all lanes contain roughly equivalent amounts of parasite protein. The robust suppression of SmCA did not result in any detectable change in adult parasite morphology or behavior in vitro. However, live worms whose SmCA expression was suppressed by RNAi treatment, unlike controls (treated either

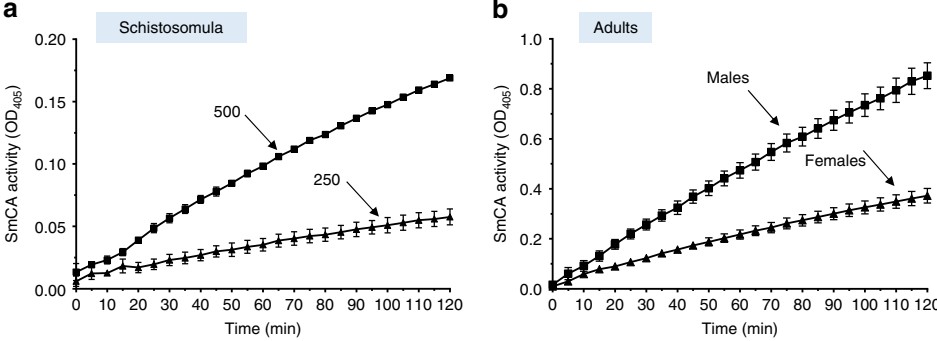

**Fig. 2** SmCA activity (mean *p*-NPA cleavage, OD$_{405}$ ± SD) observed in live schistosomula (three groups of 250 or 500, as indicated) over time (**a**) or in live, individual male (*n* = 8) or female (*n* = 12) adult worms, as indicated (**b**)

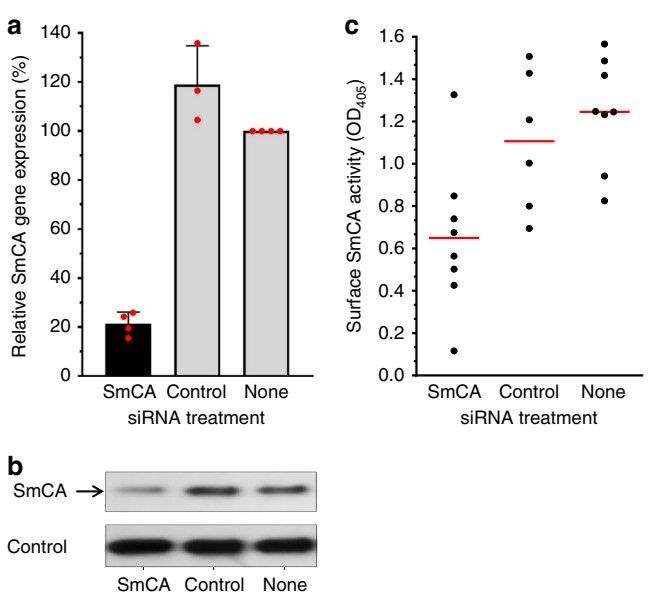

**Fig. 3** Suppression of SmCA using RNAi. **a** Relative SmCA gene expression (mean ± SE) in adult parasites 2 days after electroporation with SmCA siRNA (black bar), control siRNA or no siRNA (grey bars, as indicated). Red symbols represent source data (i.e., mean values from 3–4 biological replicates). **b** SmCA protein expression analysis. SmCA protein detected in adult worm extracts obtained 7 days following treatment with the indicated siRNAs. Notably less SmCA is seen in extracts from parasites treated with SmCA siRNA (left lane, arrow) compared to the extracts of parasites treated with control siRNA (center) or with no siRNA (None, right panel). The lower panel shows a strip of the gel stained with an irrelevant antibody that yields strong staining in all lanes to illustrate roughly equivalent protein loadings in each. **c** SmCA enzyme activity (p-NPA cleavage, OD$_{405}$) in individual live adult male worms 7 days after treatment with either SmCA siRNA, irrelevant (Control) siRNA or no siRNA (None). The red lines indicate the mean values for each group. Mean surface SmCA activity for the SmCA-suppressed group is significantly lower than that of either control group (*P* < 0.05, *n* = 6–8)

with the irrelevant, control siRNA or with no siRNA), had a significantly diminished ability to cleave the exogenously added, synthetic CA substrate, p-NPA, *P* < 0.05, (Fig. 3c).

Next, we investigated whether RNAi-mediated gene silencing of SmCA had any impact on the parasites in vivo. To do this, 1-day cultured schistosomula were first electroporated with either an siRNA targeting the SmCA gene or with a control, irrelevant siRNA. The parasites were then cultured for 4 days before

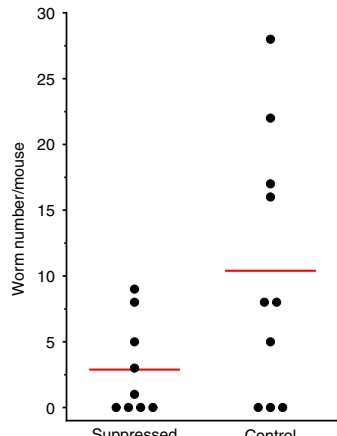

**Fig. 4** Schistosome recovery from infected mice following SmCA suppression. Schistosomula were injected i.m. into mice on day 4 after treatment with SmCA siRNA or control siRNA. Mice were perfused 42 days later, and worm burdens measured. Each dot represents the worm burden from a single mouse and the red lines indicate the means for each group. Mean worm recovery from the SmCA-suppressed group is significantly lower than that of the control group (*P* < 0.05, *n* = 9–10)

introducing them into mice, as described in Methods. Infected mice were perfused 42 days later. The number of worms recovered was counted and data are presented in Fig. 4. Each data point represents the number of worms from an individual mouse and the red lines represent the mean for the group. Following the control treatment, a mean of 10.4 (±3.1) worms was recovered and this is significantly more than the number recovered from the suppressed group (2.9±1.2 worms; *p* < 0.01); following SmCA gene suppression there was a 72% reduction in the mean worm burden compared to control.

**Expression of recombinant SmCA**. To express SmCA, pSec-Tag2A plasmid encoding a his-tagged/myc-tagged, secreted form of SmCA was first transfected into CHO-S cells. After ~48 h, the culture supernatant was collected and recombinant SmCA (rSmCA) was purified using standard immobilized metal affinity chromatography (IMAC). In lane 1, Fig. 5a, the purified rSmCA, running at ~50 kDa, is seen in this Coomassie Blue stained gel (arrow) alongside molecular mass markers (lane M). This band stains strongly with anti-SmCA antibody (α-SmCA) as assessed by western blotted (Fig. 5a, lane 3). An extract of adult parasites, resolved by SDS-PAGE, blotted to PVDF membrane and similarly probed with α-SmCA, reveals clear staining of a sharp band —native SmCA—also running at 50 kDa (arrow, Fig. 5a, lane 4).

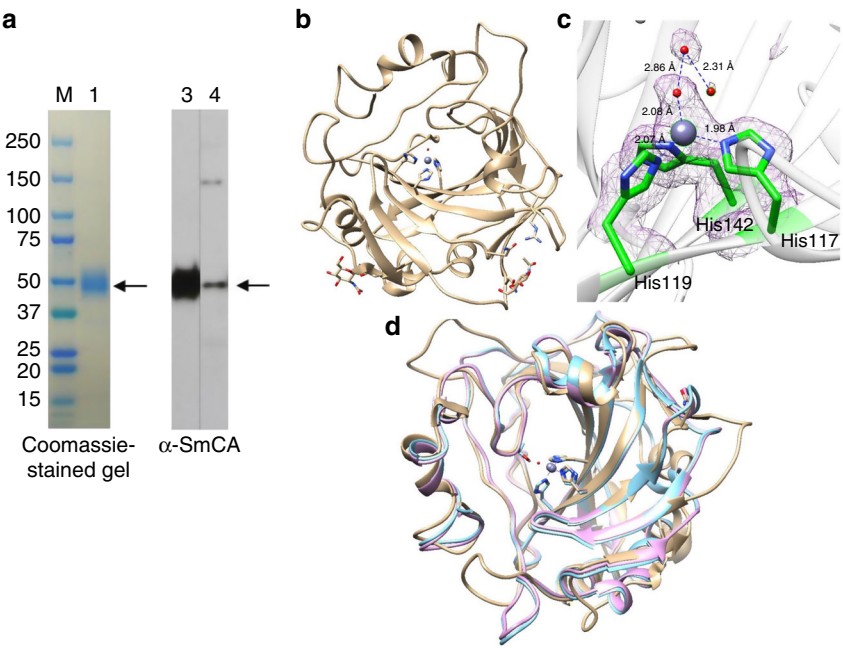

**Fig. 5 a** Purification of rSmCA by immobilized metal affinity chromatography (IMAC). Lane 1 shows purified rSmCA after resolution by SDS-PAGE and stained with Coomassie Blue. The arrow shows rSmCA. M represents molecular size markers (kDa). Anti-SmCA antibodies detect rSmCA (lane 3), and native SmCA in an adult schistosome extract (lane 4, arrow), by western blot analysis. **b–d** Crystal structure of rSmCA. Ribbon diagram of the SmCA monomer structure (**b**) along with depiction of the enzyme active site in a space fill and stick model (**c**) with the zinc ion (gray sphere), its ligands (His[117], His[119], and His[142], indicated) and water molecules (red spheres). A Fo–Fc difference Fourier omit map is also shown contoured at 2 σ. Panel **d** represents superposition of a monomer of SmCA (tan) with the two hCAs II mutants (PDB 4PXX, pink and PDB 4HBA, light blue, whose sequences are given in supplementary fig. 4). The catalytic zinc ion is represented as a gray sphere

Recombinant SmCA activity was measured using the $CO_2$ hydration reaction: The $k_{cat}$ of SmCA in this reaction is $1.2 \times 10^6$ per second and $k_{cat}/K_m$ is $1.3 \times 10^8\,M^{-1}.S^{-1}$.

Since the molecular weight of SmCA (~50 kDa) is greater than the predicted size of the protein based on its amino acid sequence (37 kDa) this is suggestive of post-translational modification. Since analysis of the SmCA sequence suggests the presence of several N-glycosylation sites, we examined the glycosylation status of the protein. Supplementary fig. 3A compares rSmCA with the native schistosome protein by western blotting both before (−) and after (+) treatment with the de-glycosylating enzyme peptide N-glycosidase F (PNGase F). In all cases, PNGase F treatment results in a marked shift in the staining pattern of SmCA. The ~50 kDa protein runs at ~37 kDa following deglycosylation. This is true for rSmCA as assessed using an anti-myc tag antibody (α-myc) or using anti-SmCA antibody (α-SmCA). This is also true for the native protein; following incubation of an adult worm extract with PNGase F a prominent band at ~37 kDa band appears. Note that PNGase F treatment clearly does not result in the complete deglycosylation of the native protein since the ~50 kDa band is clearly detected both before and after treatment (supplementary fig. 3A).

**Crystal structure of SmCA**. The rSmCA structure was determined by molecular replacement using the coordinates of a highly thermally stable variant of human CA II (PDB entry 4PXX) and refined to 1.75 Å resolution in the P3₂21 space group (Table 1, Methods). The maps showed clear density for the majority of residues, thus enabling modeling of the structure, with the exception of roughly the first 20 amino acids (i.e., the signal peptide) and the last 25 amino acids, which were not included in the model. Figure 5b shows a ribbon diagram of the SmCA monomer structure (top, left) along with depiction of the enzyme

active site (top right) in a space fill and stick model with the zinc ion (gray sphere), its ligands (His[117], His[119], and His[142], indicated) and water molecules (red balls). The SmCA structure shares with all other α-CAs the same tertiary fold, with a central ten-stranded twisted beta-sheet as the dominant secondary structure element. Among the CAs of known structure, the highest level of homology (35% identity) was found with two mutants of hCA II (PDB: 4PXX and 4HBA). Fig. 5d represents superposition of a monomer of SmCA (tan) with the two hCAs II mutants (PDB 4PXX, pink and PDB 4HBA, light blue, whose sequences are given in supplementary fig. 4). This structural comparison shows that while the overall SmCA structure is similar to that of hCA II, a number of differences exist: First, SmCA is ~20 residues longer at the N-terminus and C-terminus. (These residues are not included in our SmCA model, see above.) Second, some connections between strands (e.g., residues 73–81, 174–182, and 283–290, SmCA numbering) are longer compared to the equivalent connections in the hCA II structure and third, SmCA residues 103–110 have a different conformation. The superposition of the SmCA model with the hCA II structure (PDB entry 4PPX) yielded a root-mean-square deviation between Cα atoms of 1.287 Å. Our analysis also confirms that SmCA is a glycoprotein. The protein contains five potential N-glycosylation sites (consensus sequence N-X-T/S) and density maps indicate glycosylation with two saccharides being found covalently bound to Asn[74] and Asn[189] of both subunits (supplementary fig. 3B, C). The corresponding electron densities were modeled to an N-acetyl-D-glucosamine residue.

As in all other α-CAs, the active site of SmCA is centered on the catalytic zinc ion (gray sphere in Fig. 5b–d), which is coordinated by three histidine residues (His[117], His[119], His[142], green/blue in Fig. 5c) and, as a fourth ligand, a water molecule in an almost tetrahedral geometry (Fig. 5c). Moreover, two other water molecules are hydrogen-bonded to the metal-bound water

**Table 1 Data collection and atomic model refinement statistics**

| | PDB: 6QQM |
|---|---|
| *Data collection* | |
| Space group | P3$_2$21 |
| Cell dimensions | |
| a, b, c (Å) | 103.04, 103.04, 132.56 |
| a, b, g (°) | 90.0, 90.0, 120.0 |
| Resolution (Å) | 39.60 -1.70 (1.81 – 1.70) |
| R$_{sym}$ (%) | 11.0 (192.8) |
| R$_{meas}$ (%) | 11.4 (200.9) |
| I / sI | 12.00 (1.00) |
| Completeness (%) | 98.7 (92.5) |
| Redundancy | 13.6 (12.46) |
| CC (1/2) | 0.998 (0.606) |
| *Refinement* | |
| Resolution (Å) | 39.60 – 1.75 |
| No. reflections | 77640 |
| R$_{work}$ / R$_{free}$ | 4278 |
| Rfactor (%) | 17.01 |
| Rfree(%) | 20.39 |
| No. atoms | |
| Protein | 552 |
| Ligand/ion | 2 |
| Water | 418 |
| B-factors | |
| Protein | 26.70 |
| Ligand/ion | 20.99 |
| Water | 33.76 |
| R.m.s. deviations | |
| Bond lengths (Å) | 0.0214 |
| Bond angles (°) | 2.283 |

Values in parentheses are for the highest resolution shell
*r.m.s* root-mean-square

**Table 2 Comparative inhibition data of sulfonamides AAZ–HCT against rSmCA**

K$_i$ (nM)

| Compound | hCAI | hCAII | SmCA |
|---|---|---|---|
| AAZ | 250 | 12 | 42.5 |
| MZA | 50 | 14 | 38.9 |
| EZA | 25 | 8 | 8.7 |
| DZA | 50,000 | 8 | 21.2 |
| BRZ | 45,000 | 3 | 46.6 |
| BZA | 15 | 9 | 34.1 |
| TPM | 250 | 10 | 568 |
| ZNS | 56 | 35 | 937 |
| SLP | 1200 | 40 | 8470 |
| IND | 31 | 15 | 36.5 |
| VLX | 54,000 | 43 | 431 |
| CLX | 50,000 | 21 | 4970 |
| SLT | 374 | 9 | 3560 |
| SAC | 18,540 | 5959 | >50,000 |
| HCT | 328 | 290 | 399 |

K$_i$ values obtained using the stopped-flow CO$_2$ hydrase assay; data for recombinant human isoforms hCAI and hCAII from[64]. Errors are in the range of ±10 % of the reported data (from three different assays). Full names of each tested compound are given in Methods

molecule, forming a well-ordered network inside the active site. The position of the water molecules (red spheres in Fig. 5c) inside the network is similar with respect to hCA II, but in SmCA a further water molecule is present close to the metal ion, which is hydrogen-bonded to the zinc-bound water (Fig. 5c).

Crystal structure analysis confirms that SmCA active site residues are well conserved with respect to hCA II. Residues that in hCA II are known to play crucial roles inside the catalytic mechanism—such as His[64], Glu[106], Thr[199], Thr[200]—are conserved in SmCA. In particular, the proton shuttle residue His[64] (His[88] in SmCA), responsible for transferring protons between the zinc-bound water and the external buffer, is found in one conformation, whereas in hCA II it is often in two different conformations. Differences between the active site of SmCA compared with hCA II include the replacement of Glu[100] in SmCA with Asn[67] in hCA II and, in the upper part of the catalytic cavity, Pro[163] and Ile[167] replace Phe[131] and Val[135]. These residues have been found to be important in the process of inhibitor adaptation to the hCA II active site and may be key in the selective targeting of SmCA for inhibition. Finally, unlike hCA II, crystal structure analysis reveals that in the SmCA structure a disulfide bond is present between Cys[45] and Cys[235].

**SmCA inhibitor screen**. In this work, the ability of a total of 52 different compounds to inhibit rSmCA activity was measured using the stopped-flow CO$_2$ hydrase assay. The first series comprised 15 compounds, including the classical CA inhibitor acetazolamide and related compounds; the K$_i$ values of each for SmCA are given in Table 2 in comparison to K$_i$ values obtained using two widely expressed human isoforms (hCA I and hCA II). The second series comprises 37 different potential anion inhibitors of rSmCA: the K$_i$ values for these are listed in Table 3.

In most cases the compounds do not show appreciably higher affinity for SmCA compared to the human isoforms. However, some of the anions exhibited substantially greater affinity for the schistosome enzyme versus the human CAs tested. For instance, both phenylarsonic acid (PAA) and phenylbaronic acid (PBA) have >1000-fold higher affinity for SmCA versus human isoforms hCA I or hCA II; the K$_i$ of PAA for SmCA is 0.029 mM versus 31.7 mM (for hCA I) and 49.2 mM (for hCA II) and the K$_i$ of PBA for SmCA is 0.02 mM versus 58.6 mM (for hCA I) and 23.1 mM (for hCA II). Sulfamide exhibits 44-fold greater potency over SmCA (K$_i$ = 0.007 mM) versus hCA I (K$_i$ = 0.31 mM) and 161-fold over hCA II (K$_i$ = 1.13 mM).

## Discussion

Schistosomes are parasitic worms of major global health importance. Following infection of a vertebrate host, the parasites undergo a remarkable biochemical and morphological transformation to adapt to internal environmental conditions. Part of this transformation involves the generation of a new syncytial surface covering called the tegument, which is bounded at the host-parasite interface by an unusual double-lipid bilayer membrane[17,36]. Recent proteomic analysis of this structure has helped enumerate its molecular composition[16]. Among the proteins identified by proteomic analysis of the *Schistosoma mansoni* tegument is a member of the carbonic anhydrase family[18,19] and it is this molecule, designated SmCA, that is the focus of our work here. Using the published proteomic data, we identified the full-length cDNA encoding SmCA and its gene. The predicted 323-amino acid SmCA protein has the features of functional enzymes belonging to the α-CA family, including conserved catalytic residues and metal binding sites. In addition, the protein has a predicted signal peptide and a conserved arginine residue at position 295, which serves as a likely GPI-anchoring site. Experimental data suggest that SmCA is indeed GPI-linked at the parasite surface since PIPLC treatment of live worms releases this protein[19]. SmCA has high sequence similarity to homologs from the two other schistosome species of great importance to human health, *S. haematobium* (ShCA) and *S. japonicum* (SjCA). The

**Table 3 Comparative inhibition data of selected, potential anion inhibitors of rSmCA**

Ki [mM]

| Compound | hCA I | hCA II | SmCA |
|---|---|---|---|
| $F^-$ | >300 | >300 | 6.44 |
| $Cl^-$ | 6 | 200 | 9.45 |
| $Br-$ | 4 | 63 | 9.35 |
| $I^-$ | 0.3 | 26 | 5.11 |
| $CNO^-$ | 0.0007 | 0.03 | 4.18 |
| $SCN-$ | 0.2 | 1.60 | 3.15 |
| $CN^-$ | 0.0005 | 0.02 | 2.05 |
| $N_3^-$ | 0.0012 | 1.51 | 2.43 |
| $HCO_3-$ | 12 | 85 | 5.10 |
| $CO_3^-$ | 15 | 73 | 8.86 |
| $NO_3-$ | 7 | 35 | 7.87 |
| $NO_2-$ | 8.4 | 63 | 5.56 |
| $HS^-$ | 0.0006 | 0.04 | 5.96 |
| $HSO_3^-$ | 18 | 89 | >100 |
| $SnO_3^{2-}$ | 0.57 | 0.83 | 4.11 |
| $SeO_4^{2-}$ | 118 | 112 | 6.33 |
| $TeO_4^{2-}$ | 0.66 | 0.92 | >100 |
| $OsO_5^{2-}$ | 0.92 | 0.95 | 7.55 |
| $P_2O_7^{4-}$ | 25.77 | 48.50 | >100 |
| $V_2O_7^{4-}$ | 0.54 | 0.57 | 4.10 |
| $B_4O_7^{2-}$ | 0.64 | 0.95 | 8.00 |
| $ReO_4^-$ | 0.11 | 0.75 | 8.75 |
| $RuO_4^-$ | 0.101 | 0.69 | 2.07 |
| $S_2O_8^{2-}$ | 0.107 | 0.084 | 7.85 |
| $SeCN^-$ | 0.085 | 0.086 | 2.16 |
| $CS_3^{2-}$ | 0.0087 | 0.0088 | 6.16 |
| $Et_2NCS_2^-$ | 0.0008 | 0.0031 | 0.59 |
| $CF_3SO_3^-$ | nt | nt | 5.62 |
| $SO_4^{2-}$ | 63 | >200 | 6.86 |
| $ClO_4^-$ | >200 | >200 | >100 |
| $BF_4-$ | >200 | >200 | >100 |
| $FSO_3^-$ | 0.79 | 0.46 | 6.02 |
| $NH(SO_3)^{2-}$ | 0.31 | 0.76 | 7.59 |
| $H_2SO_3H$ | 0.021 | 0.39 | 0.017 |
| **$H_2NSO_2NH_2$** | **0.31** | **1.13** | **0.007** |
| **$Ph-B(OH)_2$** | **58.6** | **23.1** | **0.020** |
| **$Ph-AsO_3H_2$** | **31.7** | **49.2** | **0.029** |

$K_i$ values obtained using the stopped-flow $CO_2$ hydrase assay; data for recombinant human isoforms hCAI and hCAII from[64]. Errors are in the range of 3–5% of the reported data (from three different assays); bold text indicates the three compounds (sulfamide, phenylboronic acid, and phenylarsonic acid) with highest relative affinity for SmCA v the human isoforms
*nt* not tested

protein also displays higher sequence conservation with the GPI-linked human CA hCA IV compared to CAs from *C. elegans*, *D. melanogaster* or to other human isoforms. While there are at least seven distinct CA families (designated α, β, γ, δ, ζ η, and θ[24]), the families displaying little amino acid sequence similarity and SmCA clearly belongs with the α-CA family. Members of this family are additionally found in vertebrates as well as many plants, algae, and some bacteria.

The fact that both rSmCA and the native enzyme both resolve by SDS-PAGE at a higher molecular mass than predicted is indicative of post-translational modification of the protein. SmCA has several predicted N-glycosylation sites and since deglycosylation of the SmCA samples using PNGaseF leads to the generation of molecular forms that now resolve at about the expected molecular mass this proves that SmCA is a glycoprotein. Indeed, further confirming this, crystal structure determinations identified carbohydrate covalently bound to Asn[74] and Asn[189].

The SmCA gene is expressed in all schistosome life stages studied, with highest relative expression in adult males, as assessed by RT-qPCR. Immunolocalization analysis shows strong expression of the SmCA protein in the tegument of all stages examined—cultured schistosomula and adult males and females. Localization of SmCA by immunogold electron microscopy confirms that the protein is distributed on the host-interactive tegumental membranes. These localization data are consistent with the protein playing a role in the blood stream of the inter-mammalian host and at the host-parasite interface. In agreement with the supposition that SmCA is not only found in the tegument but is also specifically located on its external surface is the finding that intact live parasites (schistosomula as well as adult male and female worms) all display an esterase enzymatic activity attributable to CAs[27,28]. This activity is noteworthy since, over 40 years ago, non-specific esterase activity was reported in the tegument and cytons of *S. mansoni* adults using histochemical techniques[29]. We hypothesize that SmCA is responsible for this activity.

To investigate the importance of SmCA for schistosomes, the expression of its gene was suppressed using RNAi. Introducing siRNAs targeting SmCA by electroporation resulted in robust gene suppression. Substantial suppression was observed at the protein level too as determined by western blotting analysis and by comparative esterase activity assays. This experiment proves that the esterase activity measured is due to the action of SmCA and is not due (at least to any great extent) to the action of other tegumental enzymes[17,37]. Living parasites whose SmCA gene had been suppressed were significantly impaired in their ability to cleave exogenously added p-NPA compared to control parasites. However, suppressing SmCA gene expression did not result in any obvious morphological or behavioral changes in the suppressed versus control parasites leading to the conclusion that SmCA does not play an important role for the worms in culture.

In order to test the effects of knocking down the expression of the SmCA gene in vivo, RNAi-treated schistosomula were cultured for 4 days after siRNA treatment before they were introduced into mice. The mice were perfused 42 days later. A significant difference in worm numbers recovered in the suppressed versus control group was observed. Approximately three-fold fewer parasites were recovered from the SmCA-suppressed group compared to the control group. Thus, the SmCA-suppressed schistosomula were significantly impaired in their ability to establish a robust infection in mice. This demonstrates that SmCA provides an essential function for schistosomes in vivo and contributes to parasite virulence. Since one important role of CAs is to maintain acid-base balance[38], parasites whose SmCA gene is suppressed may be impaired in their ability to control the pH of their immediate environment. Such changes could block the function of other schistosome tegumental enzymes, several of which act optimally at alkaline pH[39–41] and are hypothesized to be key in the ability of the worms to impair host purinergic signaling pathways and thus to impede protective immune responses[42]. Carbonic anhydrases are also essential for carbon dioxide transport out of tissues[38] so worms with suppressed SmCA gene expression may be unable to efficiently remove $CO_2$. In other systems, elevated $CO_2$ levels cause mitochondrial dysfunction, block cell proliferation[43] and have been reported to exert considerable adverse effects on the free-living worm, *Caenorhabditis elegans*[44], but the precise molecular mechanisms involved are unclear. The biochemical effects of SmCA knockdown might exert little adverse impact on worms in their relatively cossetted environment in culture but could serve to make the local environment of the worms toxic in vivo thus impairing their ability to survive inside the vasculature.

Given that SmCA gene knockdown impairs parasite survival, it seems reasonable to speculate that drugs that block SmCA action and mimic the RNAi effect should likewise debilitate the worms. Carbonic anhydrases are druggable targets[30,31] and many such

enzymes from vertebrates, fungi, protozoa, bacteria, as well as other parasitic worms, are well-known drug targets[32–35,45,46]. The exposed nature of SmCA—expressed on the external surface of intravascular schistosomes as shown here—adds to its attraction as an accessible new drug target.

Therefore, we first generated recombinant SmCA in CHO-S cells. Purified rSmCA was functional, exhibiting both esterase activity (i.e., cleavage of p-NPA) as well as $CO_2$ hydration activity (i.e., conversion of $CO_2$ to bicarbonate and protons). The latter is considered the physiologic reaction and here SmCA exhibits exceptional catalytic activity; it displays kinetic parameters approaching those of human isoform II (hCA II)—one of the best catalysts known in nature.

The crystal structure of rSmCA was determined by molecular replacement at 1.75 Å resolution, which revealed the same tertiary fold as in all other known α-CAs. The active site of SmCA does not differ substantially from that of other α-CAs and all of the important active site residues are conserved. It is noteworthy that SmCA possesses a disulfide bond between $C^{45}$ and $C^{235}$; while almost all bacterial α-CAs have a disulfide bond in the same position[47], among the human CA isoforms, only those that are membrane-bound (hCA IV, hCA IX, hCA XII, and hCA XIV) or secreted (hCA VI) contain this disulfide bond and SmCA is bound to the external parasite membrane[48]. Despite the global similarity in structure of SmCA to other members of this protein family, several sizable differences were noted between the crystal structure of SmCA versus hCA II, for instance in terms of regional conformation, in the length of connections between strands and in active site configuration and these distinctions may be key in efforts to identify specific SmCA inhibitors.

Two sets of compounds were tested for their ability of impede rSmCA action—a collection of sulfonamides and a set of potential anion inhibitors. Carbonic anhydrases are classically inhibited by compounds with a sulfonamide-based ($SO_2NH_2$) zinc-binding group or their bioisosteres (sulfamates and sulfamides). Sulfonamides bind in a tetrahedral geometry, interacting directly with the catalytic zinc in their deprotonated form, and inhibit CA activity by displacing the zinc-bound water/hydroxide ion. Metal complexing anions represent a second class of CA inhibitors[49]. Here we seek anion or sulfonamide chemicals that show superior inhibition of the schistosome enzyme compared to the human isoforms tested in this work (hCA I and hCA II). In most cases the $K_i$ values obtained with SmCA were greater than or equal to those exhibited by one or other of the hCAs, making these chemicals poor candidates for specific anti-SmCA action. However, both phenylarsonic acid (PAA) and phenylbaronic acid (PBA) exhibited over 1000-fold greater binding affinity to SmCA versus human isoforms hCA I or hCA II. Sulfamide too exhibited considerably greater affinity (44 to 160-fold) over SmCA versus the hCAs. These data show that it is possible to obtain selective inhibitors of SmCA that could act as lead compounds for the development of new drugs to block essential schistosome metabolism, blunt parasite virulence and so debilitate these important global pathogens.

## Methods

**Parasites and mice**. The Puerto Rican strain of *Schistosoma mansoni* was used. Adult male and female parasites were recovered by perfusion from female 6–8-week-old Swiss Webster mice that were infected with ~100 cercariae, 7 weeks previously[50]. Schistosomula were prepared from cercariae released from infected snails. All parasites were cultured in DMEM/F12 medium supplemented with 10% heat-inactivated fetal bovine serum, 200 µg/mL streptomycin, 200 U/mL penicillin, 1 µM serotonin, 0.2 µM Triiodo-l-thyronine, 8 µg/mL human insulin and were maintained at 37 °C, in an atmosphere of 5% $CO_2$[41]. Parasite eggs were isolated from infected mouse liver tissue[50]. All protocols involving animals were approved by the Institutional Animal Care and Use Committees (IACUC) of Tufts University.

**Cloning SmCA**. Proteomic analysis of the *S. mansoni* tegument revealed a potential partial carbonic anhydrase homolog[18,19] (accession number Smp_168730). BLAST interrogation of the *S. mansoni* genome (version 3) with this sequence led to the identification of the likely SmCA gene. Using oligonucleotides designed from the predicted 5' UTR just upstream of a predicted initiator methionine (SmCAF: 5'-AACATTAAATAATCCAATTTAT -3') and the 3'UTR downstream of the last predicted exon (SmCAR: 5'- AAGAATTTCGGTCTAGAAGAAGAAGC -3'), with adult parasite cDNA in a PCR, we amplified and then sequenced, at the Tufts University Core Facility, the complete SmCA coding DNA.

**Anti-SmCA antibody production**. The following peptide comprising SmCA amino acid residues 72–94 was synthesized: NH2-YRNTS-STETTTIQNNGHSAEVKFC-COOH by Genemed Synthesis, Inc. San Antonio, TX. A cysteine residue was added at the carboxyl terminus to facilitate conjugation of the peptide to bovine serum albumin (BSA). Approximately 500 µg of the peptide-BSA conjugate in Freund's Complete Adjuvant was used to immunize a New Zealand White rabbit subcutaneously. The rabbit was boosted with 100 µg of peptide alone in Incomplete Freund's Adjuvant 20, 40, and 60 days later. Ten days following this, serum was recovered and anti-SmCA antibodies were affinity-purified using a peptide-ovalbumin conjugate and dialyzed against phosphate buffered saline (PBS).

**SmCA gene expression analysis**. The levels of expression of the SmCA gene in different life stages of the parasite, and in parasites treated with gene-specific siRNA, was measured by reverse transcription quantitative PCR (RT-qPCR), using a custom TaqMan gene expression system (Applied Biosystems). RNA was isolated from parasites using the TRIzol reagent per the manufacturer's guidelines (Thermo Fisher Scientific). Residual DNA was removed by DNase digestion using a TurboDNA-free kit (Applied Biosystems). cDNA was synthesized using 1 µg RNA, an oligo (dT)20 primer and Superscript III RT (Thermo Fisher Scientific). The levels of expression of the SmCA gene in different life stages was measured by RT-qPCR using the housekeeping gene triose phosphate isomerase as the endogenous control[51]. Primer sets and reporter probes labeled with 6-carboxyfluorescein (FAM), obtained from Applied Biosystems were used for RT-qPCR. To detect SmCA expression, the following primers and probe were used: SmCA-F: 5'-GTGGATCTGAGCATACAATTGATGGA -3'; SmCA-R: 5'- CACTGGGTGAA GAATACATCTGTCTT -3' and probe SmCA-M2: 5'-FAM-AGATTTCCTTTA GAAGGACAT -3'. Each RT-qPCR was performed using a TaqMan PCR mix, cDNA and primer and probes in a final volume of 20 µL. All samples were run in triplicate and underwent 40 amplification cycles on a Step One Plus Real Time PCR System Instrument. For relative quantification, the ΔΔCt method was employed[52] and the schistosome tubulin gene was used as the within-stage, endogenous control[53].

**Western blot analysis**. To monitor the expression of the SmCA protein, samples were first homogenized in ice-cold 0.1% Triton X-100 in PBS and protein content was measured using the BCA Protein Assay Kit (Pierce) according to the manufacturer's instructions. Five micrograms of protein from each sample were resolved by SDS-PAGE under reducing conditions and blotted to PVDF membrane. This was then incubated in a 5% skim milk in Tris-buffered saline (TBS) solution containing 0.1% Tween 20 (TBST) for 1 h at room temperature. The membrane was next probed overnight at 4 °C with affinity purified, rabbit anti-SmCA antibody at 1:1000 dilution. After washing the membrane three times in TBST, bound primary antibody was detected using horseradish peroxidase-labeled anti-rabbit IgG (1:5000, GE Healthcare). The blots were developed using ECL Detection Reagents (GE Healthcare) according to the manufacturer's instructions and images were recorded using a ChemiDoc™ Imaging System (Bio-Rad). To monitor protein loading per lane, the blot was first stripped of bound SmCA antibody using Restore Stripping Buffer according to the manufacturer's instruction (Thermo Scientific) and was then re-probed with control antibody.

**SmCA activity measurement**. To measure SmCA activity in live parasites, we exploit the esterase activity displayed by enzymes in this class whereby para-nitrophenyl acetate (p-NPA) is converted into para-nitrophenol (which is detected at $OD_{405}$)[28]. Groups of 7-day cultured schistosomula, or individual adult male or female worms, were incubated in assay buffer (25 mM Tris–Sulfate buffer, pH 7.4, 75 mM NaCl, 20 mM glucose) containing 1 mM p-NPA substrate, (Sigma–Aldrich), and changes in optical density were monitored over 60 min at 405 nm using a Synergy HT spectrophotometer (Bio-Tek Instruments). Trace background activity readings were subtracted from all data.

SmCA catalyzed $CO_2$ hydration activity was assessed in 10 mM HEPES (pH 7.5), 0.1 M $NaClO_4$ buffer for 10 s at 25 °C. An applied photophysics stopped-flow instrument was used with $CO_2$ concentrations ranging from 1.7 to 17 mM using Phenol red (0.2 mM) as indicator and at the absorbance maximum of 557 nm, as previously described[35].

**SmCA immunolocalization**. Adult worms were either recovered following perfusion of infected mice or remained in situ within a blood vessel that was dissected from a 7-week infected mouse. These tissues were embedded in Optimal Cutting

Temperature (OCT) compound (Thermo Fisher Scientific) and flash frozen in liquid nitrogen. Sections (7 μm thick) were then prepared using a cryostat, applied to slides and fixed in cold acetone. Immunofluorescent detection of SmCA was carried out using affinity-purified, rabbit anti-SmCA antibody diluted 1:100 and Alexa fluor 488-conjugated goat anti-rabbit IgG (Thermo Fisher Scientific), essentially as described earlier[54]. Control parasite sections were treated with Alexa fluor 488-conjugated secondary antibody alone. Some sections were additionally incubated with DAPI (4',6-diamidino-2-phenylindole, Thermo Fisher Scientific) for 5 min and Phalloidin (Thermo Fisher Scientific) for 60 min at room temperature to stain DNA and actin, respectively. The same protocol was used to assess the localization of SmCA in whole 3-day cultured schistosomula, which were fixed in cold acetone.

**Immunogold labeling and electron microscopy**. Freshly perfused adult parasites were fixed overnight with 2% glutaraldehyde in 0.1 M cacodylate buffer at 4 °C. The samples were then dehydrated in a graded series of ethanol, then infiltrated and embedded in L.R. white acrylic resin. Ultramicrotomy was performed using a Leica Ultracut R ultramicrotome and the sections collected on gold grids. Grids were immunolabeled in a two-step method according to the following procedure; the grids were conditioned in PBS for 5 min × 3 at room temperature, followed by the blocking of non-specific labeling for 30 min at room temperature using 5% non-fat dry milk in PBS. After rinsing, the grids were exposed to primary anti-SmCA antibody diluted 1:30 for 1 h at room temperature, followed by washing in PBS and then incubated with secondary antibody diluted 1:30 (10 nm gold-labeled goat anti-rabbit IgG (H&L, GE Healthcare)) for 1 h at room temperature, and finally rinsed thoroughly in water. The grids were exposed to osmium vapor and/or lightly stained with lead citrate to improve contrast and were examined and photographed using a Philips CM 10 electron microscope at 80 KV.

**Suppression of SmCA expression using RNAi**. Worms were treated either with a synthetic siRNA targeting SmCA (SmCA siRNA1: 5'- CTAA-CAACTCCACCATGCACAGAAA -3') or with control siRNA, which targets no sequence in the schistosome genome (5'-CTTCCTCTCTTTCTCTCCCTTGTGTA-3'). Delivery of siRNAs to the parasites was by electroporation as described previously, using 3 μM of each siRNA in electroporation buffer (BioRad)[55]. Gene suppression was assessed post-treatment by comparing mRNA levels (using RT-qPCR) and protein levels (by western blotting analysis and enzyme activity measurements) in target versus control groups, as described above.

**Infection of mice with siRNA-treated schistosomula**. One-day-old cultured schistosomula were electroporated with either SmCA siRNA or control siRNA. Parasites were then maintained in culture for 4 days before being washed and counted and used to infect female Swiss Webster mice by injecting ~1000 schistosomula in 100 μl of RPMI (without phenol red) into the thigh muscle of each animal using a 1 mL tuberculin syringe and a 25G-1 needle[56]. Worms were recovered from all mice by vascular perfusion 6 weeks later and counted.

**Expression and purification of recombinant SmCA**. The full-length coding sequence of SmCA (GenBank accession number, MK611932), including the predicted signal peptide and GPI anchor domain, was codon optimized using hamster codon preferences and synthesized commercially (Genscript). Next, the region encoding amino acids $N^{21}$–$A^{298}$ (i.e., lacking the amino terminal signal peptide and the carboxyl terminal GPI-anchoring signal) was generated by PCR using forward and reverse primers containing *Asc*I and *Xho*I restriction sites, respectively, and the synthetic codon-optimized gene as template. The amplified product was cloned into the pSecTag2A plasmid (Invitrogen) at the *Asc*I and *Xho*I sites in frame with the Igκ leader sequence at the 5'-end and a myc epitope and 6-histidine tag at the 3'-end. Successful in-frame cloning was confirmed by sequencing at the Tufts University Core Facility.

To express recombinant SmCA (rSmCA), suspension-adapted FreeStyle Chinese Hamster Ovary Cells (CHO-S) were transfected with plasmid using Free-Style Max Reagent following the manufacturer's instructions (Invitrogen). Cells were harvested at various time points post-transfection to monitor viability (by Trypan Blue exclusion test) and rSmCA expression (by western blotting). To facilitate protein production, stable cell line clones secreting rSmCA were selected by treating transfected cells with 250 μg/mL Zeocin for 2 weeks; individual clones that produced high yields (5–10 mg) of purified active rSmCA/L were maintained.

Recombinant SmCA was purified from cell culture medium by standard Immobilized Metal Affinity Chromatography (IMAC) using HisTrap™ Excel columns, following the manufacturer's instructions (GE Healthcare Life Sciences). Purified recombinant protein, eluted from the column, was dialyzed overnight at 4 °C against 50 mM Tris–HCl (pH 7.4), 150 mM NaCl, then concentrated by ultrafiltration centrifugation (Pierce Protein Concentrators, 10 K MWCO, Thermo Scientific). Final protein concentration was determined using a BCA Protein Assay Kit (Pierce). Aliquots of eluted protein were resolved by 4–20% SDS-PAGE to assess purity and some were tested for specificity by western blotting using anti-myc tag and anti-SmCA antibodies.

**SmCA crystallization and data collection**. The enzyme was crystallized at 296 K using the sitting-drop vapor-diffusion method in 96-well plates (CrystalQuick, Greiner Bio-One). Drops were prepared using 1 μL protein solution mixed with 1 μL reservoir solution and were equilibrated against 100 μL precipitant solution. The concentration of the protein was 10 mg mL$^{-1}$ in 50 mM Tris, pH 8.3. Initial crystallization conditions were found using the JCSG-plus screen (Molecular Dimensions) and were optimized. Diffraction-quality crystals grew within 4 days to approximate dimensions of $0.2 \times 0.2 \times 0.3$ mm from a solution consisting of 20% PEG 3350, 0.2 M $KNO_3$. The crystals belonged to the primitive trigonal space group P $3_221$, with unit-cell parameters $a = 103.04$, $c = 132.56$ Å. A native data set extending to a maximum resolution of 1.75 Å was collected on the BM29 beamline at ESRF (Grenoble, France) using a DECTRIS Pilatus 6M-F detector and a wavelength of 1.0399 Å. For data collection, a crystal of the enzyme was cooled to 100 K using a solution consisting of 20% PEG 3350, 0.2 M Potassium nitrate and 15% glycerol as cryoprotectant. The data were processed with XDS[57].

The structure was solved by the molecular-replacement technique using MOLREP[58] with the coordinates of the structure of a highly thermally stable variant of human CA II (PDB entry 4PXX) as a starting model. The model was refined using REFMAC5[59] from the CCP4 suite[60]. Manual rebuilding of the model was performed using Coot[61]. Solvent molecules were introduced automatically using ARP/wARP[62]. Refinement resulted in R-factor and R-free values of 0.1707 and 0.2034, respectively. Data processing and refinement statistics are summarized in Table 1. The Ramachandran plot is of high quality; 96.13% of the non-glycine and non-proline residues are in the most favored regions and 3.87% are in the additionally allowed regions. Protein coordinates have been deposited in the Protein Data Bank (PDB entry 6QQM). Structural figures were generated with the UCSF Chimera program[63].

**SmCA inhibitor screens**. All compounds were obtained commercially (Sigma–Aldrich) and exhibited >95% purity, as assessed by HPLC. Tested compounds include several that are in use clinically: acetazolamide (AAZ), methazolamide (MZA), ethoxzolamide (EZA), dorzolamide (DZA), brinzolamide (BRZ), benzolamide (BZA), topiramate (TPM), zonisamide (ZNS), sulpiride (SLP), indisulam (IND), valdecoxib (VLX), celecoxib (CLX), sulthiame (SLT) saccharin (SAC), and hydrochlorothiazide (HCT). Anions tested in this study included physiological ones, such as $Cl^-$ and $HCO_3^-$, as well as metal complexing anions, such as halides and pseudohalides (iodide, bromide, azide, cyanide, cyanate, and thiocyanate), hydrosulfide, and arsenate, together with fluoride, nitrate and perchlorate.

$K_i$ values for each compound were obtained using the stopped-flow $CO_2$ hydrase assay above. At least six traces of the initial 5–10% of the reaction were used to determine initial velocity. Uncatalyzed rates were determined in the same manner and subtracted from the total observed rates. Stock solutions of inhibitors were prepared in distilled-deionized water, and dilutions up to 1 nM were made thereafter with assay buffer. Enzyme and inhibitor solutions were pre-incubated together for 15 min before assay, to allow for the formation of any enzyme–inhibitor complex. Potential anion inhibitors were added to the assay as sodium salts, except sulfamide, phenylboronic acid, and phenylarsonic acid. Inhibition constants were obtained by non-linear least-squares methods using PRISM 3 and the Cheng–Prusoff equation, as previously described[64–66].

**Statistics and reproducibility**. For data generated by RT-qPCR one-way analysis of variance (ANOVA) and Tukey as the post-hoc analysis was used. Measurements were taken from distinct samples. To analyze the live worm enzyme activity data, two-way repeated measures ANOVA was used. To assess worm recovery one-way ANOVA was used. In all cases, $P$-values <0.05 were considered significant.

**Reporting summary**. Further information on research design is available in the Nature Research Reporting Summary linked to this article.

## Data availability
All data supporting this study are presented here (or as supplementary information). Plasmids and antibodies are available upon request from the corresponding author, P.J.S.

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

## Acknowledgements

This work was supported by grant AI-056273 and AI-111011 from the National Institutes of Health—National Institute of Allergy and Infectious Diseases (NIH-NIAID). Schistosome infected snails were provided by the Biomedical Research Institute through NIH-NIAID contract HHSN272201000009I. We thank John Nunnari for help with electron microscopy.

## Author contributions

P.J.S. and A.A.D. conceived the project; A.A.D. characterized SmCA on worms, identified its biological significance, and generated rSmCA; A.A., M.F., and C.T.S. crystalized SmCA and undertook chemical inhibitor screens; P.J.S. directed the project and assembled the manuscript; all authors edited the manuscript.

## Additional information

**Competing interests:** The authors declare no competing interests.

