## [Peer Review File · Communications Biology]

Reviewers' comments:

Reviewer #1 (Remarks to the Author):

The report of Da'dara and collaborators describes the crystal structure of *S. mansoni* carbonic anhydrase together with immunolocalization of the protein in the tegument, kinetic parameters determination and small screening inhibitor assay. The finds of the paper are novel and provides a new approach to fight the Schistosomiasis.

This is well a written and well conduct report. The SmCA was expressed in mammalian cells showing glycosylation state of the protein, probably for reduce the immunological host response.

The worm recovery after RNAi treatment was reduced by 72% demonstrating the huge importance of SmAC for the worm. This is the strong part of the manuscript, however is also the weakest point, the authors does not explore the importance of the presence of SmAC in the worm tegument for the worm metabolism. What the consequence for the worm the reduction of SmAC in the tegument? Which pathways could be compromised without SmAC? Host-parasite immunological interactions? Worm metabolic problems? The authors must include the physiological relevance of the SmAC for the worm.

Supplementary table 1.

Please insert Rpim values. Due high redundancy of the data, both Rsym and Rmeas is no long appropriate, you could choose maintain or remove these parameters. For modern data collection and processing I recommend the paper "How best to use photons" of Winter and collaborators. Acta Cryst. (2019). D75, 242-261.

Both rmsd deviations of angles and bonds are high, this is probably a consequence of using Refmac. I suggest using Phenix in the next refinements.

Reviewer #2 (Remarks to the Author):

"Crystal structure and chemical inhibition of an essential schistosome host-interactive virulence factor – carbonic anhydrase SmCA" characterizes the *S. mansoni* gene product of Smp_168730 and postulates its role as a virulence factor through its carbonic anhydrase activity. The authors show that SmCA is localized to the tegument and provide evidence that it is likely anchored to the outer surface and in contact with the host environment. Enzyme activity was tested for both native surface SmCA on parasites and recombinantly expressed enzyme and furthermore, recombinant enzyme was successfully crystallized resulting in a structure. Suppression of Smp_168730 using RNAi treatment was used to examine the effect on virulence. Finally, a selection of potential inhibitors was screened against recombinant SmCA.

Overall, the manuscript is well-written and is technically sound. It establishes SmCA as a possible drug target for the neglected tropical disease, schistosomiasis, that is currently treated worldwide using a single drug. Importantly, while the SmCA structure is very similar to the human enzyme structures, the inhibitor screen results indicate that enough differences exist to possibly develop selective inhibitors for schistosomal tegument-bound carbonic anhydrase. A few points to consider are listed below:

1) The control to indicate background activity is not shown in the SmCA activity panels in Figure 2. If background activity was subtracted from the data instead, it should be indicated in the text.

2) Recovery of worms from mice infected with siRNA-treated schistosomula shown in Figure 4 appears

to be inefficient as infection with a large number of worms showed at most a recovery of ~30 in the control set. Additionally, 3 of 10 control mice showed no worm burden. Please comment on the nature of the experiment regarding the inefficiency.

3) It is not necessary to show the asymmetric unit view in Figure 5B. The structural alignment in Supplementary Figure 4B (top) would be better placed in the main text in Figure 5B after the single molecule view. Only one view of the alignment is sufficient. Also, add root mean square deviation comparisons of the SmCA structure to the human enzyme structure coordinates to the results section.

4) Page 12, line 250: correct to "corresponding electron densities"

5) Page 22, line 481 and Page 28, line 599: end these sentences "as previously described."

Reviewer #1

The worm recovery after RNAi treatment was reduced by 72% demonstrating the huge importance of SmAC for the worm. This is the strong part of the manuscript, however is also the weakest point, the authors does not explore the importance of the presence of SmAC in the worm tegument for the worm metabolism. What the consequence for the worm the reduction of SmAC in the tegument? Which pathways could be compromised without SmAC? Host-parasite immunological interactions? Worm metabolic problems? The authors must include the physiological relevance of the SmAC for the worm.

The reviewed has identified an important and central point of our paper: SmCA gene knockdown has no overt impact on worms *in vitro* but severely limits their ability to survive *in vivo*. Like the reviewer, we are keenly interested in knowing the key function of SmCA that mediates this effect and we speculate in the text as follows (lines 361 - 374):

*Since one important role of CAs is to maintain acid-base balance³⁸, parasites whose SmCA gene is suppressed may be impaired in their ability to control the pH of their immediate environment. Such changes could block the function of other schistosome tegumental enzymes, several of which act optimally at alkaline pH^{39, 40, 41} and are hypothesized to be key in the ability of the worms to impair host purinergic signaling pathways and thus to impede protective immune responses⁴². Carbonic anhydrases are also essential for carbon dioxide transport out of tissues³⁸ so worms with suppressed SmCA gene expression may be unable to efficiently remove CO₂. In other systems, elevated CO₂ levels cause mitochondrial dysfunction, block cell proliferation⁴³ and have been reported to exert considerable adverse effects on the free-living worm, *Caenorhabditis elegans*⁴⁴, but the precise molecular mechanisms involved are unclear. The biochemical effects of SmCA knockdown might exert little adverse impact on schistosomes in their relatively cossetted environment in culture but could serve to make the local environment of the worms toxic *in vivo* thus impairing their ability to survive inside the vasculature.*

In the absence of additional hard data, we are reluctant to speculate further in this manuscript on the molecular function of SmCA.

Supplementary table 1.

Please insert Rpim values. Due high redundancy of the data, both Rsym and Rmeas is no long appropriate, you could choose maintain or remove these parameters. For modern data collection and processing I recommend the paper "How best to use photons" of Winter and collaborators. Acta Cryst. (2019). D75, 242-261. Both rmsd deviations of angles and bonds are high, this is probably a consequence of using Refmac. I suggest using Phenix in the next refinements.

We have added the Rpim values and removed Rsym and Rmeas values; we thank the reviewer for the paper recommendation and the suggestion regarding the use of Phenix in future.

Reviewer #2:

1) The control to indicate background activity is not shown in the SmCA activity panels in Figure 2. If background activity was subtracted from the data instead, it should be indicated in the text.

We have added a sentence to the manuscript to the effect that “Trace background activity readings were subtracted from all data.” (line 490).

2) Recovery of worms from mice infected with siRNA-treated schistosomula shown in Figure 4 appears to be inefficient as infection with a large number of worms showed at most a recovery of ~30 in the control set. Additionally, 3 of 10 control mice showed no worm burden. Please comment on the nature of the experiment regarding the inefficiency.

As the reviewer points out, schistosome infection of experimental animals is indeed inefficient; intramuscular injection of experimentally derived schistosomula (as used in our work) is the most efficient of several modes of infection previously examined. (See: James ER and Taylor MG (1976) *Journal of Helminthology* 50, 223—233; *Transformation of cercariae to schistosomula: A quantitative comparison of transformation techniques and of infectivity by different injection routes of the organisms produced.*) Low recovery of parasites even under the control infection condition (as seen by us) is the norm for such experiments. (See e.g. Tran MH, et al. (2010) *Suppression of mRNAs encoding tegument tetraspanins from Schistosoma mansoni results in impaired tegument turnover.* PLoS Pathog 6(4): e1000840.)

3) It is not necessary to show the asymmetric unit view in Figure 5B. The structural alignment in Supplementary Figure 4B (top) would be better placed in the main text in Figure 5B after the single molecule view. Only one view of the alignment is sufficient. Also, add root mean square deviation comparisons of the SmCA structure to the human enzyme structure coordinates to the results section.

We have rearranged the figures as suggested by the reviewer and we have added to the results section the root mean square deviation comparisons as requested (line 250).

4) Page 12, line 250: correct to “corresponding electron densities”

With thanks, we have made the correction. (Now line 255.)

5) Page 22, line 481 and Page 28, line 599: end these sentences “as previously described”.

We have added the suggested text. (Now lines 496 and 614.)

Changes to the manuscript are highlighted. We hope that the revised work is now suitable for publication in *Communications Biology*.

REVIEWERS' COMMENTS:

Reviewer #1 (Remarks to the Author):

The authors answered satisfactorily all the questions raised. I recommend the publication in the current format, with one minor correction.

1. The CC (1/2) value in Supplementary Table 1 should be 0.998 (0.606) instead 99.8 (60.6).

Reviewer #3 (Remarks to the Author):

The authors have addressed the comments and publication in Communications Biology is recommended.

We have complied with referee #1's request that "The CC (1/2) value in Supplementary Table 1 should be 0.998 (0.606) instead 99.8 (60.6)." Note that, as per Communications Biology instructions, this table is now incorporated in the main manuscript file as table 1.